# Co-Learning Computational and Design Thinking Using Educational Robotics: A Case of Primary School Learners in Namibia

**DOI:** 10.3390/s22218169

**Published:** 2022-10-25

**Authors:** Annastasia Shipepe, Lannie Uwu-Khaeb, Carmen De Villiers, Ilkka Jormanainen, Erkki Sutinen

**Affiliations:** 1School of Computing, University of Eastern Finland, 80100 Joensuu, Finland; 2Department of Computing, Mathematical & Statistical Sciences, University of Namibia, Windhoek 10026, Namibia; 3Department of Computing, University of Turku, 20014 Turku, Finland; 4Nakayale Private Academy, Ruacana 16006, Namibia

**Keywords:** computational thinking, design thinking, educational robotics, programming, co-learning, primary school, Namibia

## Abstract

In a two-day educational robotics workshop in a Namibian primary boarding school, learners with no programming skills managed to apply both computational and design thinking skills with the aid of educational robotics. Educational robotics has proved to be an area which enhances learning both computational thinking and design thinking. An educational robotics (ER) workshop focusing on Arduino robotics technologies was conducted with primary school learners at Nakayale Private Academy. Observation methods through watching, listening and video recordings were used to observe and analyze how the learners were interacting throughout the workshop. Based on the results, it was concluded that this approach could be applied in classrooms to enable the primary school learners apply computational and design thinking in preparation of becoming the producers and not only the consumers of the 4IR technologies.

## 1. Introduction

The fourth industrial revolution (4IR) is here with the cutting edge and emerging technologies in areas such as robotics and it is shaping societies and industries across the world [1,2]. It is therefore about time that belts are tightened and to ensure the present and future generations in Namibia, Africa and the world at large understand and apply the robotics technologies that revolve around the 4IR. Robotics is an area under the 4IR that is picking up speed worldwide; however, this area is underrepresented in Namibia. It is vital that learners in Namibia understand how robotics technologies can be applied from an early stage through educational robotics. Educational Robotics (ER) involves the teaching of how robotics applications and systems can be designed, analyzed and operated [3]. For children, getting to know robotics at an early stage prepares them for the future and for the opportunities that robotics bring about because this will make them the producers of robotics technologies and not only the consumers of robotics products [4,5].

Robotics is an interdisciplinary field which involves programming, electronics and other fields such as engineering [6,7]. Programming is a field that involves writing scripts to develop software applications as well as writing scripts that enable certain hardware components to communicate [8]. The programming module has been one of the most feared modules at the university level and it is evident that this could be because of the conservative, teacher-centered, outdated and examination-oriented teaching approaches and methods used at the universities [9].

Electronics, on the other hand, is defined as the branch of physics and technology that involves the design of circuits using transistors, microchips, and other components [7]. Anyone who starts working with robotics must understand the basics of electronics for safety reasons of both the robotic components and the human beings. In general, ER activities and initiatives have increased entering schools in the past years, and this could be because ER activities are believed to trigger motivation in learners and students which then stimulates the learning process [3,10]. An educational robotics workshop was given to grade 6 and 7 learners at Nakayale Private Academy, a charity school located in Etunda area near Ruacana, in Omusati region deep down in the Northern part of Namibia. The aim of this workshop was to equip learners with robotics skills, and this involved the learners’ first-time exposure to both text-based and visual programming. The learners also got to apply the loop structure well, given the fact that they acquired knowledge on programming for the first time at the workshop and this is discussed later in this paper.

Gyebi, Hanheide and Cielniak argue that ER plays a vital role in tackling some challenges that African institutions of higher learning face, in a paper where he addressed an open question on the efficacy of ER on students’ performance and experience [11]. They found that the integration of educational robotics in undergraduate curriculum does not only equip students with robotics skills but also provides them with skills that they need to build the real robotics systems. The aim of this study was to explore how primary school learners with no knowledge on programming and robotics can be exposed to design and computational thinking using Arduino technologies through an ER workshop. This was accompanied by learners applying the 4Cs of 21st century skills, which are communication, collaboration, critical thinking and creativity.

The next section of this paper talks about what scholars say about computational thinking, design thinking, imagination & creativity, robotics technologies and curricula. The paper further talks about the methods used to conduct this workshop and make the learners apply both robotics technologies and programming because this was their first time working with real Arduino robotics components. This section is then followed by the results, and then discussions and conclusions.

## 2. Literature Review

### 2.1. Computational Thinking (CT) and Design Thinking (DT)

Design thinking and computational thinking are two vital ways of interpreting how humans approach design problems [12]. DT and CT are considered to be a dual-process model used in providing solutions to design problem [12]. The two forms of thinking are said to be mirrors to each other in which thinkers envision the solution to the design problem by moving between the two processes. Both DT and CT are considered to have a positive impact in addressing problems and they are being employed by schools and universities although they are in most cases taught in isolation of another [12,13].

Nowadays, many international educational policies emphasize that children should be equipped with computational thinking skills from preschool [14]. Computational thinking includes four main elements to solve or provide solutions to problems, and these are decomposition, pattern recognition, abstraction and algorithms [15,16]. CT has its roots in the science of how solutions to design challenges can be provided using computers [12,17]. Yildiz and Süleyman defined computational thinking as an area that involves problem solving skills in a way that both the computer and human can understand [14]. They further added that CT is rated as one of the competence areas in acquiring computer programming nowadays. It has been reported by scholars that various countries have already incorporated computational thinking into their K-12 curriculum and there have been requests to make computational thinking compulsory for learners of all level [12]. It is stated by Kaleliogˇlu and Gülbahar that computer programming plays a vital role in gaining problem solving skills [18].

Programming requires not only logical thinking but design thinking also needs to be taken into consideration when teaching students to solve real-life problems [19]. DT has its roots in the science of how people act, think and reason when addressing design problems [12]. The definition of design thinking varies from scholar to scholar, for example Zheng defines it as a way of solving problems using design philosophy centered around stakeholders, abductive reasoning, visualization, holistic perspective and experimentation [20].

According to various scholars, design thinking skills are mostly taught through project-based learning and phenomenon-based learning in which the learners are required to apply universal thinking skills to solve problems [17,21]. Nakata and Hwang define design thinking as a way of solving human centered problems using a design-based approach [21].

Tech giants such as Google, Apple and Samsung use design thinking when creating new products putting consumers at the center of the designing process and ensuring that they design products that the users will be interested in using [22]. The design thinking process is said to have five steps that need to be taken into consideration and the steps are empathize, define, ideate, prototype and test. Some of these steps were well followed during the robotics workshop when the learners were solving the tasks assigned to them. Computational thinking and design thinking are summarized in Table 1.

### 2.2. Imagination and Creativity

Imagination and creativity have been used in middle school to have the learners understand the content being taught in an imaginary yet creative way [23]. Imagination, also referred to as educational imagination is connected to realization and remembrance. It has been argued in research that a human being would hardly bring things to reality if they are unable to apply imagination. Creativity or creative thinking in the educational context is a process in which learners not only construct new ideas but also revamp existing ideas to come up with new various innovative ideas. Moreover, some scholars define creative as a process of ideation whose results is satisfies the peers involved.

A model of imagination and creativity in academic learning (ICAL) was introduced in a study conducted by Beghetto and Schuh [24]. This ICAL model was built on the previous model based on creative learning and knowledge construction. The main aim of the ICAL model is to table the alliance of imagination and creativity in academic learning [24]. The ICAL model is said to have two main components which are educational imagination (EI) and educational creativity (EC) operating under intrapsychological and interpsychological spheres. The first component is the EI which enables learners to envisage and visualize newly introduced concepts in curriculum. This is usually made possible by the learner’s previous knowledge and learning experiences and trajectories which then enables the learners to easily formulate new ideas. Educational creativity is the second ICAL model whose definition has its roots in the ideation and design process birthing unique products in academic learning [24]. EC provides distinctive, effective, useful and unique ways of meeting academic needs in the contexts of learners’ interpretation of the content being presented to them.

As mentioned above, educational imagination and creativity operate at both intrapsy-chological and interpsychological spheres of which intrapsychological focuses on personal and individual where else intrapsychological focuses on social aspects.

In short, the educational imagination and creativity are the key to the initiation and creation of possibilities. These two components of ICAL model were used and applied in the two-day ER workshop as explained in the results section.

### 2.3. Tinkercad as Both Visual and Text-Based Programming Tool

Tinkercad is a web-based platform that exposes students to both design and computational thinking skills through the use of 3D design, circuits design and code block menus [25]. A study was carried out in Turkey to measure the perceptions of students on Tinkercad and it was concluded that Tinkercad as a platform is easy to use and highly useful [25,26]. It is also reported by Eryilmaz and Deniz that Tinkercad was developed to help individuals become the producers of technology and not only the consumers of technology [25]. This is through individuals learning and applying 3D designs using online tools, instead of them learning on real 3D printing machines that are quite expensive for students to afford.

Tinkercad mainly allows learners and students to gain circuit design and 3D design skills by designing their own designs on the platform depending on the problem they intend to solve such as Arduino based problems [27]. The latter is linked to learners and students creating block-based code and text-based code in Tinkercad allowing them to apply algorithmic thinking, critical thinking, collaboration and creativity, as well as problem solving skills [25].

In recent years, increasingly, block-based visual programming tools with different infrastructures have been developed [28]. Generally, these environments are designed to appeal to individuals of all ages with no programming experience [18,28]. The latter is one of the reasons why text-based integrated development environments were given a thought of being eliminated [14], but maybe the elimination of text-based could be for children. It is seen in the literature that learning programming with visual tools has a lot of positive outcomes especially for children [14]. Visual programming tools play an important role in learning and teaching robotics and it is clear in the literature that robotics can be used to support learning in the field of Science, Technology, Engineering, Arts and Mathematics (STEAM) [5,29,30]. STEAM involves the designing and creation products using a combination of science, technology, engineering, arts and mathematics fields [5].

### 2.4. Arduino/Lego and Children in Africa

AfrikaBot is South Africa’s first competition on robotics and it was initiated mainly for African students to learn robotics technologies [31]. AfrikaBot enables teenagers to gain technological and entrepreneurial skills especially those from low-income households and they achieve this by providing teenagers with a built-it-yourself robot post the robotics competition [31]. Researchers from AfrikaBot worked tirelessly for eight years to create a robotics community by creating awareness through their website, YouTube channel and by providing affordable robotics kits for the competition.

A RobotScience project was also initiated by Coach Michael Ettershank with the aim of teaching young South Africans programming and robotics [32]. The kids at RobotScience learn how to create and develop robotics systems using Lego but it is also reported that Coach Michael has developed a low-priced Arduino hardware platform. Among others, MindsInAction in Namibia equips grades 4 to 12 learners with robotics skills and sparks interest in the STEAM field in them by exploring robotics, mechatronics, coding and electronics [9]. Scholars argue that countries need to prepare for continual and unforeseen shifts in technological and economical shifts by centering their educational systems around STEAM for all the subjects/courses regardless of the educational level, be it primary, secondary or tertiary level [5].

Physically Active Youth (P.A.Y.) in Namibia also hopes to contribute to the students’ life-skills and educational success by implementing a robotics program [33]. Physically Active Youth (PAY) is one of the after-school programs whose aim is to equip the youth with 21st century skills through robotics [33]. PAY operates at the Multipurpose Youth Centre, located in Katutura, Windhoek, Namibia. PAY was mainly started to focus on project-based learning as opposed to theoretical traditional teaching and learning systems that is still very common in Namibia [33]. The theoretical traditional teaching and learning systems is, however, now being addressed in Namibia with project-based learning picking up and being incorporated in the educational systems especially for higher education and for some primary schools [33]. An educational robotics program gives P.A.Y students the opportunity to develop their problem-solving skills and confidence to enter STEAM fields [33].

Robotschool is also one of the after-school programs in Windhoek that equip students with the stem skills through robotics using the project-based learning approach [33].

### 2.5. ER Curricula at Elementary School Level

A study was conducted by Chaudhary et al., to explore the effectiveness and application of Lego Mindstorms EV3 to teach programming coupled with teamwork, project management and problem-solving to primary school level kids [4]. During this study, nine classes with lots of hands-on practices were conducted at a summer camp to equip the learners with the aforementioned skills. It was reported from observation that learning with robots makes learning fun and enables students to tackle even challenging problems in a fun yet productive way [4]. It was further concluded that teaching and learning with robots where learners are able to design and connect robotic components is more effective as opposed to lecture-theory based approach.

A robotics programming course was also designed by Noh and Lee in Turkey, targeting elementary school learners and they investigated the impact of the course by teaching it in real classrooms [14]. They implemented the course using grade 5 and 6 learners and their study concluded that teaching programming with the aid of robots really improves the learners’ creativity and numerical thinking. Although they observed that computational thinking skills did not really pick up among learners during this study, the same approach could be applied in the same context to teach visual programming to learners at elementary schools with the aid of robotics. According to Noa and Lee, visual programming courses have been incorporated in many curricula of various countries around the world [14].

Teaching robotics to primary school kids, i.e., 4th–7th grade is becoming popular throughout the world as robotics is now part of most of the school curriculum; however; this is different in the Namibian context as the current curriculum does not include robotics at all. Based on the official syllabus documents available on National Institute for Educational Development (NIED)’s website, the current Namibian curriculum/syllabus for primary school does not include programming nor robotics [34]. NIED is a directorate that was established in 1990 within the Ministry of Education, Arts and Culture (MoEAC) to tackle the curriculum design and development for basic education in Namibia [34].

The integration of ER and CT in the school curriculum is not only an educational matter of concern in Namibia alone but also worldwide [35]. A recent study was undertaken in Barcelona, Spain looking at the integration of CT and ER into their secondary school curriculum using a project-based learning approach with its focus on the development of STEAM skills [35].

### 2.6. Programming with Children in Africa

It is important to acquire programming skills from an early age because it helps them develop computational thinking skills, creativity, logical thinking, algorithmic thinking, problem solving and innovation skills [36]. Having said that, robotics is an area that de- pends heavily on programming because almost every robotics platform, requires coding for the robotic systems to work. It can also be said that programming can also depend on robotics because robotics aids the learners to apply the programming skills using robotics technologies and have a clear picture on how robotics and programming are related.

The loop structure is considered to be one of the challenging topics to understand in programming [37,38]. Teaching methods could be the contributing factor to this, because it is evident that some teaching pedagogies used not only in African schools but also in other parts of the world are not engaging the students [5]. The latter is the reason why selected institutions in Africa made an effort to include educational robotics in their curricula really to improve teaching and learning because ER is considered to be more effective and more engaging and triggers motivation in students [11].

In a recent article by Lamech Amugongo, he emphasized that Namibia needs to make sure that children are equipped with the necessary skills to take up space in the era of 4IR technologies [39]. He further added that coding needs to be taught to pupils at an early age because it teaches them not only problem-solving skills but also creativity.

There is an open question of how primary school learners from the marginalized community can be equipped with programming and basic robotic skills and what the impact of robotics to primary school learners are? To contribute to this body of knowledge, a two days’ robotics workshop was conducted with primary school learners at Nakayale Private Academy, in the Ruacana area, north of Namibia. This robotics workshop introduced learners not only to the basics of programming and electronics but also to programming’s loop structure. The learning pedagogies used in this workshop are unpacked in the results section of this paper. Teaching and learning with robotics are said to be highly effective when it comes to tackling some programming concepts such as loops [40].

## 3. Research Design

### 3.1. Research Problem

Namibia has recently taken a step to catch up with the 4IR technology by assessing the country’s readiness for the fourth industrial revolution and this involved discussions on upgrading the country’s education to education 4.0. Education 4.0 focuses on motivating and exciting teaching and learning pedagogies using the emerging technologies such as robotics. Research on educational robotics in Namibia is at a very early stage and still needs more exploration hence this study focused on getting an answer to the research questions below.

### 3.2. Research Questions

RQ1. How can a participant-oriented educational robotics workshop prepare primary school learners to apply 4IR technologies?

RQ2. How can education practitioners incorporate robotics, computational and design thinking in the Namibian curriculum?

### 3.3. Research Methodology

This is part of a more extensive investigation that is being guided by the Design Science Research (DSR) Methodology (shown in Figure 1), a methodological approach that focuses on the designing and development of innovative artifacts that solve societal problems; however, in this study we have only focused on designing the artifact through the observation method. The artifact in this context is the ICDC framework (presented in Section 4.4) used in the study to understand and apply the basics of robotics alongside computational thinking and design thinking. 

### 3.4. Context and Participants

The workshop was attended by 24 primary school learners from Nakayale Private Academy, a charity school in the northern part of Namibia. Nakayale Private Academy was established in January 2016 with the aim of improving the livelihood of children from the marginalized community and orphans who are being neglected by providing them with access to high quality education and providing them with literally everything from free tuition fee, basic needs, accommodation, healthcare and world class educational exposure [41]. Three numbers of groups were formed at the beginning of the workshop and the same groups were kept throughout the workshop. All the groups had equal gender representations totaling to 8 females and 16 males aged between 12 and 13.

### 3.5. Workshop Timetable

The workshop was guided by the schedule shown in Table 2.

### 3.6. Data Collection and Analysis

Data collection was performed by the researchers through observation where learners were being observed on how they were applying computational thinking to solve problems assigned to them using Arduino technologies. There was also an interview with Eagle FM, one of the radio stations in Namibia. Eagle FM called in to find out how the workshop was going and how the workshop was impacting the Namibian child.

### 3.7. Ethical Consideration

The University of Eastern Finland’s (UEF) ethical guidelines as defined by the Finnish Advisory Board on Research Integrity were followed throughout the workshop. Considering that the learners were minors, permission was granted from the school director and staff members who are also caregivers of learners at Nakayale Private Academy and an informed consent was signed. All participants were honestly informed about the aim and purpose of the workshop and that participation was voluntary. The researcher respected the autonomy of the learners throughout the workshop.

### 3.8. Workshop Facilitators

The workshop was facilitated by three facilitators, i.e. the coordinator of the Future Tech Lab of the University of Turku’s plug-in campus in Namibia; a PhD student of the University of Eastern Finland and a Professor who is also the campus director of University of Turku’s plug-in campus in Namibia. All of them are from a computer science background.

### 3.9. Robotics Technologies and Components Used

The robotics technologies and components used in the workshop are shown in Table 3.

## 4. Results

The two-day workshop was attended by 24 grade 6 and 7 learners. It kicked off with facilitators and learners introducing themselves. Four groups were then formed, and each group had six learners, with three learners sharing a laptop per group. The learners were from Nakayale Private Academy, a charity school located in Ruacana, in the northern part of Namibia. In addition to that, the learners are from the marginalized community, mostly the Himbas and Dhembas of Namibia. As mentioned above, these were primary school learners with no prior knowledge of programming and robotics and the following content unpack how these learners managed to apply the basics of robotics and programming in two days. It is important also to mention here that the schoolteachers were assigned to the workshop not only to monitor the progress of the learners but they were also learning themselves as they were following using their mobile devices and tablets.

### 4.1. Workshop Introduction

The workshop facilitators started off the workshop with an introductory activity which seemed to have created an open relationship between the facilitators and the learners. The learners were asked to introduce themselves by saying their expectations for the workshop and a fun fact about them individually. After this activity, the learners were open to ask questions and interact with the facilitators.

Part of the learners’ expectations was for them to know what robotics is. Since the learners were never taught robotics before, their expectations of robotics was made easier by first showing them different Arduino (shown in Figure 2) and LEGO (shown in Figure 3) robotics system prototypes developed by engineering students at another robotics workshop conducted at José Eduardo Dos Santos (JEDS) Campus, one of the University of Namibia’s campuses. This was mainly carried out by bringing in the context of educational imagination where learners could imagine and visualize the solutions to the tasks they were assigned to work on throughout the workshop.

The two-day workshop with the primary school learners was, however, only based on Arduino. The facilitators introduced the learners to Tinkercad for the learners to first understand electronics and the designing of Arduino robotics systems on the software before they had to try it on real Arduino equipment.

Keeping in mind that we were dealing with primary school learners, we were going through every component slowly, explaining the components to them and making sure everyone understood before moving on to the next step. The learners were then assigned tasks to assess and observe whether they really understood what they were being taught.

### 4.2. Group Tasks

Robotics is an area that involves other areas, such as coding, design, engineering and mechanics, for it to be fully operational [6,7]. It is also evident that robotics is an area that motivates teams to engage and compete in a healthy way as they learn the STEAM skills and the 21st Century skills [33]. Hence the learners at NPA were divided into three teams to easy the learning and teaching process throughout the workshop and for them to work on the assigned tasks. It was observed that the learners were free to try out new things and do things different. This is not a common practice in a Namibian culture where learners would usually follow the methods taught by teachers only.

#### 4.2.1. Task 1: Introduction to Tinkercad

The facilitators introduced the learners to Tinkercad by explaining the components in Tinkercad one by one as well as teaching them the basics of electronics by designing with them on the platform shown in Figure 4.

During this task, it was observed that the learners managed to connect the wires, dragged and dropped the LED light from the components area to the work environment (shown in Figure 5), with the aim of turning on the LED light on the Tinkercad platform. The learners were warned to be cautious and make sure they understand everything Tinkercad before they try out real Arduino components. Figure 6 shows the three teams working on task 1.

#### 4.2.2. Task 2: Turning on the LED Light on Arduino Board

As mentioned above, the learners moved on to turning on the LED on the Arduino board using the following components: jumper wires, Arduino board, LED light, battery and breadboard (as shown in Figure 7). This task was further extended to them working with 2 LEDs and then 3 LEDs. It was observed that the learners were fast at grasping the knowledge and interesting enough they were collaborating in helping each fellow learners understand specific tasks mimicking what they have designed on Tinkercad.

#### 4.2.3. Task 3: Arduino Integrated Development Environment (IDE)

Learning or teaching how the Arduino Integrated Development environment (shown in Figure 8) was a little challenging to the learners given the complexity of terms and components involved. At this stage the instructors configured the Arduino IDE for the learners and they were only told to copy and paste the code from Tinkercad platform to the Arduino IDE and execute the code. This was mainly carried out for the codes to be installed on the real Arduino board and have it produce the intended output. What was interestingly observed here was that the learners were cautious with the components when connecting the Arduino board to the laptops, keeping the basics of electronics in mind.

#### 4.2.4. Task 4: Loop Structure

It is evident that the loop structure is one of the hard to learn concepts in programming at university level [42]; however, it was interesting seeing the grade 6 and 7 learners grasping, understanding and applying the loop structure (shown in Figure 9) with the aid of robotics. One of the tasks assigned to learners was to create a robotics system with two LED lights that mimics the traffic lights behavior. The task enabled learners to work with a loop for the first time and grasp the skills fast in a motivating way.

One of the groups used the loop structure to make the LED flash on and off, by using an LED light, jumper wires, Arduino board and a breadboard. It was observed that the learners were able to explain how they achieved their goals. It was further observed that the learners were applying the concept of educational creativity in this task because they figured out other ways of connecting the components which were different from the ways of the instructors and still produced the intended task outputs. It was also amazing seeing learners at Nakayale Private Academy understanding loops within two days using Tinkercad and later practicing it on Arduino boards.

### 4.3. General Workshop Observations and Recommendation to Practitioners

In general, it was observed that the learners were afraid to ask questions in the beginning because they were unfamiliar with the facilitators as well as what they should do and might have been a bit afraid to break things. They might have also felt pressure of the teachers as the teachers were very proud to show what they have taught the learners in their normal everyday classes. Teachers at NPA were also excited to learn and see what the learners were doing as they also stood at the back but close to the kids and paid a lot of attention and learn something new (that they could also possibly use or refer to in their future lessons). Learners were also very excited to explain what they did when asked, especially when they got to understand the equipment and their functionalities. In short, they understood different topics very fast. Furthermore, when they were asked to help out fellow learners, they were very excited to do so as they could act as “the teacher” and explain it their own way which also turned out to be the way the other learner understood.

Robotics could be used in multiple other subjects in schools. It could be used to stimulate learning subjects such as mathematics and physics. It can be used in any subject depending on the teachers’ and learners’ understanding, imagination and creativity. Learners enjoy building by themselves and seeing visuals which will most likely encourage them to understand, remember and apply the topic they are being taught better.

As buying equipment might not be possible for most schools in developing countries, the teacher can still allow learners to work together to solve and explain the work to each other. If robots are not available, they can make use of any visual and physical tools to show to learners and let them interact with them.

### 4.4. ICDC Framework in the ER Workshop Context

We developed the ICDC framework in Table 4, which represents imagination, creativity, design thinking and computational thinking. The framework is summarizing how the four components were contextualized in the educational robotics workshop through observation.

### 4.5. Challenges Encountered during the Workshop

The first day was a bit challenging due to the internet connection because the data somehow became depleted as we were accessing Tinkercad, a simulator that learners were using to design the prototypes during the first day of the workshop. They needed to use Tinkercad to first prototype and test the robotics systems before they use the real Arduino equipment. The team from Windhoek had, however, a Telecom Namibia mobile internet device which was not really picking up signals and this could be because there is no transmitter nearby. The challenge was, however, solved when the NPA deputy school director called the service provider to sort out the school Wi-Fi. Another challenge faced was on the configuration of the Arduino IDE, the learners could not understand how to configure it given their educational level and the instructors had to configure the IDEs for them by going to them group by group.

### 4.6. Interview with Eagle FM

The Eagle FM radio station called in on the second day of the workshop to find what the robotics workshop was about and the progress of the workshop. They interviewed the workshop facilitators, the school’s deputy director and four of the learners that took part in the workshop. The four learners explained what they learned during the workshop and one learner further added that the workshop greatly contributed to her future career decisions. Since Eagle FM is a national radio station, the nation was also given a chance to call in and ask questions regarding the workshop as well as airing their thoughts on the educational robotics workshop. There were positive and somehow mixed feelings from the nation viewpoint. Although some people had little understanding of what educational robotics was about, the majority expressed that they were happy with what we were doing and requested that we reach out to more schools.

## 5. Discussion

In this study, primary school learners with no programming and robotics skills were equipped with the basics of robotics skills, computational skills as well as design thinking skills. The world is moving toward a digitalized world, and it is evident that young people need to be prepared for emerging 4IR technologies such as robotics. Although a lot of work on educational robotics has been carried out regionally and internationally, there are very few studies that have focused on educational robotics at primary school level in Namibia. This could be because the current primary school curriculum for public schools in Namibia does not include teaching computational and design thinking nor robotics; however, discussions have been going on to include those in the educational curriculum in Namibia. With a vast growth of emerging technologies and the world being in the midst of the 4IR, there is a need for children in Namibia and in the African continent at large to grow up with not only an understanding of using AI technologies such as robotics but with the skills and knowledge on how to apply these technologies. The latter can be achieved through educational robotics workshops such as the one conducted at Nakayale Private Academy because the approach used in the workshop through observation can be employed by educational practitioners when integrating robotics in the curriculum to prepare primary school learners to apply and even design for 4IR technologies. As per the authors’ knowledge, this robotics workshop was new in the context of Namibia’s marginalized community and it raised a lot of attention in the nation through the interview with Eagle FM radio, wanting this kind of educational robotics workshops to reach all corners of the country.

Furthermore, discussions are going on throughout the country on the integration of educational robotics and computational thinking into the curriculum of both primary and secondary school level to equip learners with skills related to 4IR technologies. Lamech Amugongo emphasized in an article [39] mentioned previously in 2.6 that Namibian children need be equipped with 4IR related skills and we believe that initiatives such as this educational robotics workshops can greatly contribute to learners being equipped with 4IR technologies. This will also help the future generations to handle explainable AI and take control of the AI machines. Amugongo also argues that coding must be compulsory for all schools in Namibia. We think that learning visual programming for robots at a primary school level ensures that learners attain coding skills because coding is an important aspect in robotics systems. It is evident that the use of block or visual programming encourages learners or students to learn and acquire knowledge on not only programming but also robotics. Having said that, a subject on visual programming at primary school level in Namibia will make it easier for learners to learn and apply robotics fast as opposed to when they learn robotics with no programming knowledge.

The flexible roles and dynamics of the workshop facilitators contributed highly to the learners being able to understand and apply the robotics technologies within two days. While one facilitator was in front teaching and explaining to the learners, the other facilitators were walking around the groups to help learners where they were struggling, and this was speaking to the concept of co-teaching. This setup and the way the workshop was conducted was completely different from the ordinal teaching pedagogy where a teacher would usually come to class and explain the subject matter without hands-on activities, leaving the learners with only theoretical knowledge and missing the practical side of the subject matter. There have been discussions in Namibia about revisiting the curriculum to include 4IR technologies. The concept of co-learning and co-teaching with hands-on activities can be key things that should be incorporated in the K-12 curriculum to ensure computational and design thinking among learners and students.

The purpose of the workshop was to explore how learners with no programming skills can be taught computational and design thinking skills through an educational robotics workshop in a way that they understand and apply robotics. Part of the purpose was also to see how this approach could possibly help the educational practitioners integrate robotics, computational thinking and design thinking in the Namibian curriculum. It can be seen in the results that the concept of design thinking played a vital role in this workshop through its empathy stage which created openness in the workshop. It was also noted that teamwork at the primary school level can help learners develop collaboration skills at a very young age and this also ensures that the class is moving together in terms of content understanding. Moreover, the concept of co-teaching also played an important role in this workshop as it was noted in the results section and this contributed to the success of the workshop. Co-teaching is, however, being practiced at university levels in Namibia, but it is not a common practice at K-12 in Namibia based on the authors’ knowledge. Hence, we recommend co-teaching and co-learning be considered in teaching subjects that require hands-on activities for K-12 education in Namibia, Africa and the world at large. It was further noted that hands-on activities especially with robotics technologies do motivate learners to learn, therefore we also recommend here that educational robotics be incorporated in Namibia’s primary school curriculum with design thinking and computational thinking.

As mentioned in the results section, this workshop also attracted Eagle FM, a local radio station which interviewed us on day two of the workshop and people who called in suggested we need to reach out to more schools. This is one of the future plans for the research group but although we can reach out to more schools we cannot reach out to all schools due to cost implications. We, however, believe that this can be achieved if robotics, computational thinking and design thinking get incorporated in Namibia’s K-12 educational curriculum.

At the end of the workshop, the learners expressed that they had an understanding of what robotics is and how it can be applied. The fact that the learners managed to grasp the basics of robotics in just two days wowed the facilitators because this was also the facilitators’ first times dealing with primary school learners with no computational skills nor robotics skills.

## 6. Conclusions

It can be concluded that the observation method used in this study could be applied in real classrooms to equip learners with robotic and other 4IR related skills. The approach used in this study would also in the meantime aid in having the learners obtain an understanding of 4IR technologies and their applications while waiting for the visual programming subject and robotics to hopefully be incorporated in the Namibian primary school curricula for both public and private schools. It is also important for teachers to create a friendly and welcoming environment in class because this encourages learners to be openly interacting in class and free to ask questions. What is important to note is that DT and ET are crucial skills in the 4IR and imagination era and many researchers have recommended them to be integrated in a wider curriculum of the educational environment i.e., from preschool to tertiary school level. Finally, the ICDC framework presented in the results section can be employed on not only ER activities but also on other handy educational activities that may include imagination, creativity, design thinking and computational thinking.

## Figures and Tables

**Figure 1 sensors-22-08169-f001:**
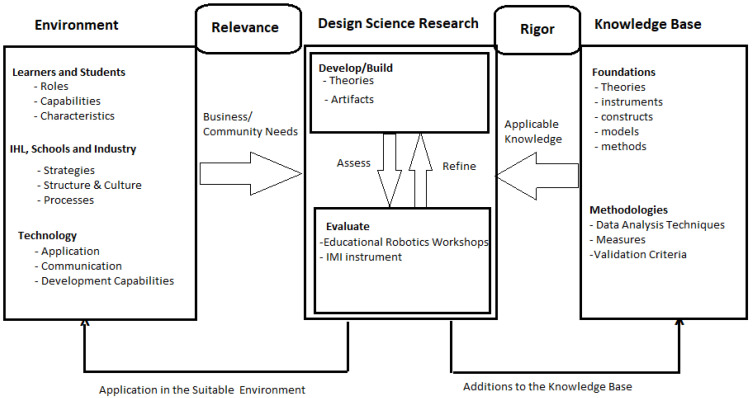
Design Science Research Methodology derived from (Hevner et al., 2004).

**Figure 2 sensors-22-08169-f002:**
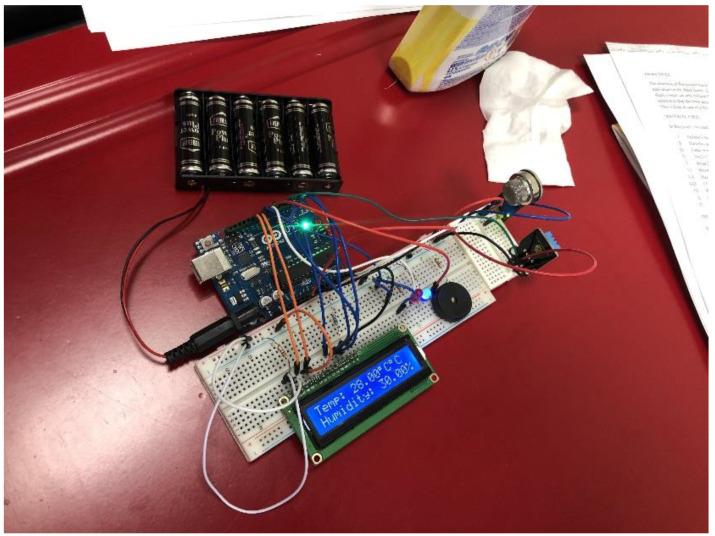
Arduino robotics prototype by engineering students at JEDS campus.

**Figure 3 sensors-22-08169-f003:**
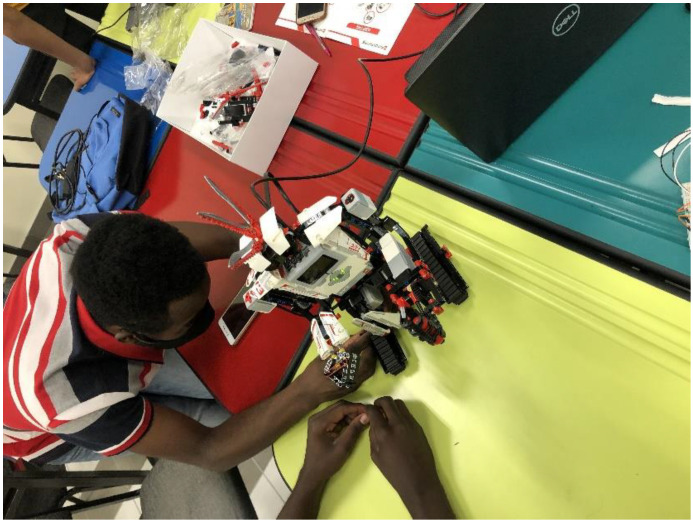
LEGO robotics prototype by engineering students at JEDS campus.

**Figure 4 sensors-22-08169-f004:**
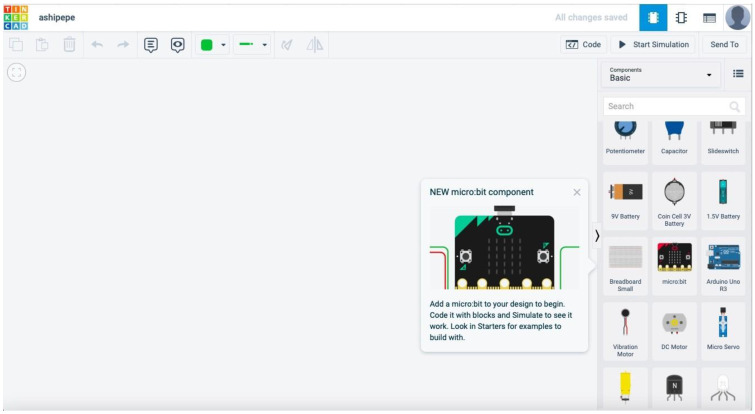
Tinkercad platform.

**Figure 5 sensors-22-08169-f005:**
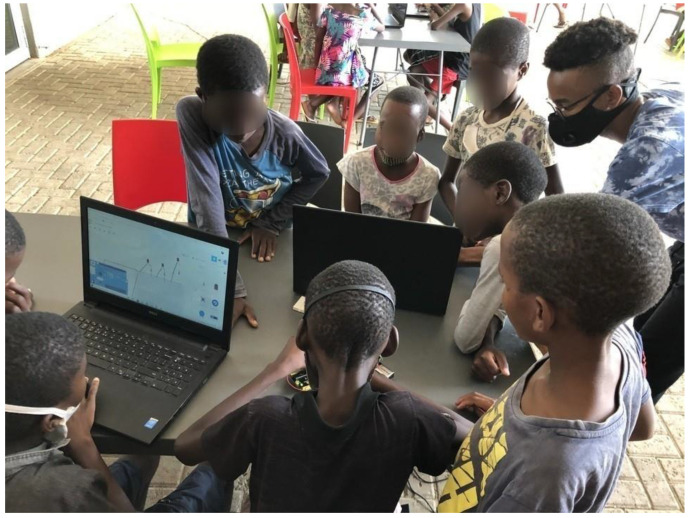
Learners working on Tinkercad.

**Figure 6 sensors-22-08169-f006:**
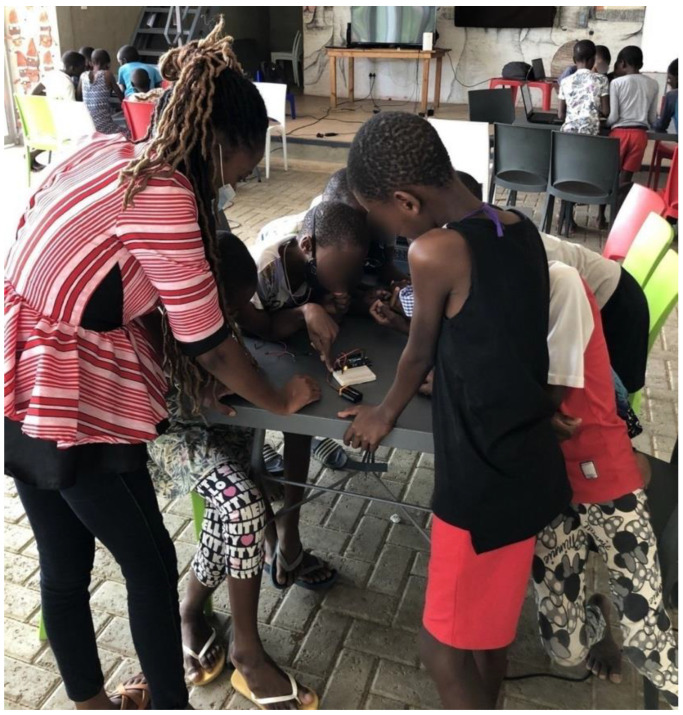
The three teams at the ER workshop.

**Figure 7 sensors-22-08169-f007:**
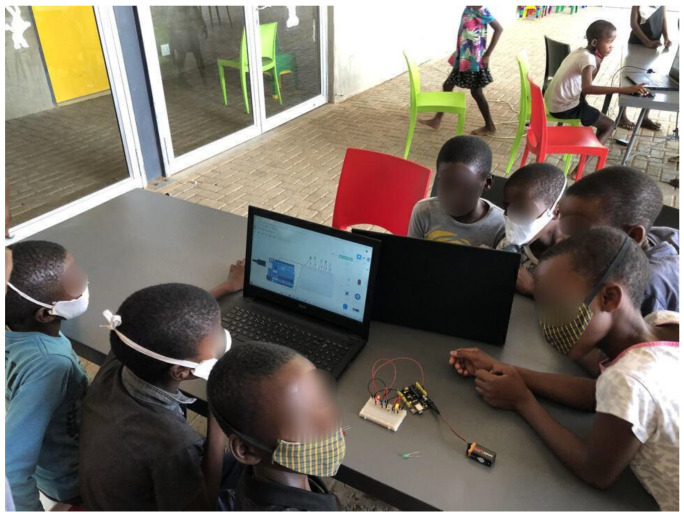
**Task 2:** LED lights.

**Figure 8 sensors-22-08169-f008:**
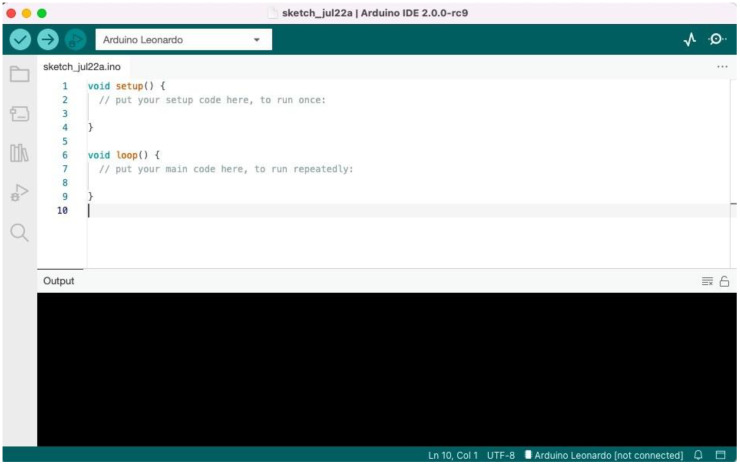
Arduino IDE.

**Figure 9 sensors-22-08169-f009:**
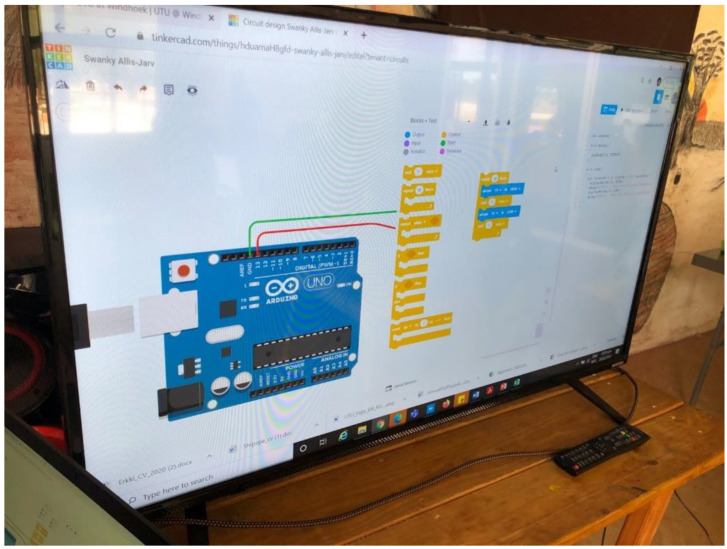
Loop structure on Tinkercad.

**Table 1 sensors-22-08169-t001:** CT and DT summary.

CT	DT	CT and DT
Focuses on the steps to provide the solution to the problem	Focuses on understanding the problem first before thinking of the solution	Both are widely popular
Influence innovation and discovery	Influence thoughtful creative problem-solving by designers	Both inspired by the design and computer science body of knowledge
Encourages students solve to unsolved problem	Developed to focus on how human beings reason in the designing process	The only forms of thinking to have gained popularity since the 21st century
Problem-solving skills and methods used by computer scientists	Has its foundation in the design cognition and design methods	
Significant to societal economic growth	Mostly used in business and government	
Involves four main elements: decomposition, pattern recognition, abstraction and algorithms	Involves five main stages: empathize, define, ideate, prototype, and test	
It depends on DT	Shapes CT	
Does not enforce that humans think like computers		

**Table 2 sensors-22-08169-t002:** The two-day workshop schedule.

Day 1	Day 2
Workshop introduction	Day 1 reflection
Basics of electronics and safety issues	Task 3: Arduino Integrated Development Environment
Task 1: Introduction to Tinkercad	Task 4: Loop structure
Task 2: Arduino actual components: Turning on LED light on Arduino board	Workshop reflection and wrap-up

**Table 3 sensors-22-08169-t003:** Technologies used.

Components		Functionalities
Arduino UNO board	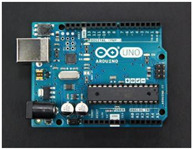	The Arduino UNO board is a microcontroller board that is based on the ATmega328P. The board has 14 digital i/o pins, and six of the 14 pins can be used as Pulse Width Modulation (PWM) outputs [42,43]. Furthermore, the board is programmable with Arduino Integrated Development Environment (IDE), using a type B USB cable [43].
Arduino IDE	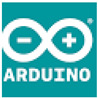	The Arduino IDE has the text editor for scripting codes which gets loaded to the Arduino board.
Tinkercad platform	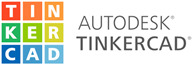	The Tinkercad platform is used to design the robotics systems.
LED lights	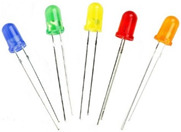	LED stands for Light Emitting Diodes. LEDs are simply powerful small lights that are used in various applications such as turning lights on and off or mimicking a traffic light [43].
Jumper wires	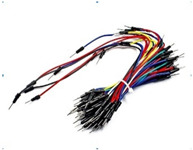	Jumper wires are used to make connections between different Arduino components [43].
Breadboard	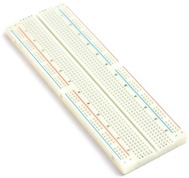	Breadboard holds electronic components such as LEDs and resistors together to allow the creation and design of easy and quick electronic circuits.
9V Lithium Batteries	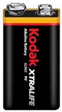	The batteries are used to provide power to the power source which supplies power to the Arduino board.
9V Battery Connector	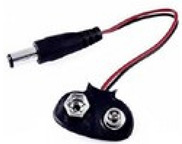	The battery connector powers the Arduino board
Resistors	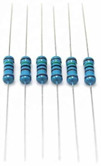	Resistors control the flow of current with the electronic circuit.

**Table 4 sensors-22-08169-t004:** ICDC Framework.

	Educational Imagination	Educational Creativity
**Design Thinking**	Facilitators imagined the workshop and showcased the previously developed robotics systems; Teachers at NPA monitoring and observing the progress of the learners in the workshop; Groups collaborating by assisting one another to solve tasks successfully; Decisions on the components needed to accomplish tasks	Creativity activities in teaching; Open interaction between participants and workshop facilitators after the introductory activity in 4.1; Creativity in designing and connecting components in Tinkercad
**Computational Thinking**	Imagining codes; Imagining a loop; Learners are assigned tasks to assess their computational skills.	Create different loops; Different ways of coding components

## Data Availability

The data reported on are available on request.

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
