# Peer review of "Co-Learning Computational and Design Thinking Using Educational Robotics: A Case of Primary School Learners in Namibia"

_sensors, 2022, doi:10.3390/s22218169_

Round 1

Reviewer 1 Report

This paper, in my opinion, is a very nice paper for any conference related with Education and Technology. But for a journal paper it needs a much more deep study and analysis of the research question; also and talking about the research question, in the abstract you talk about "Based on the results, it was concluded that this approach can be applied in classrooms to enable the primary school learners to apply computational and design thinking in preparation of becoming the producers and not only the consumers of the emerging technologies." and in the section 3.1 you say "How can primary school learners with no prior knowledge on programming and robotics be exposed to design thinking, computational thinking and educational robotics?" and finally in conlusions you say "It can be concluded that the approach used in this study to co-learn computational and design thinking using ER could be applied in real classrooms to equip learners with no prior computational skills."

In fact you have several design questions, but the study has not enough results to make such conclusions.....

You should think in sending the paper to a conferences and for having a paper for a journal you have to make a more deep study, maybe with more studentes, out of the school but also in school, to have some more feedback from the teaches, like how they have been trained, etc.....

Author Response

The abstract, research questions and conclusion were revised.

Reviewer 2 Report

In this paper learning process in a two-day educational robotics workshop in a Namibian primary boarding school were investigated where learners with no-prior programming skills managed to apply both computational and design thinking skills using educational robotics.

Based on the video recordings and on-site observation, the authors analyse how the learners were interacting throughout the workshop. And suggested that this approach can be applied in classrooms to enable the primary school learners to apply computational and design thinking in preparation of becoming the producers and not only the consumers of the emerging technologies.

The paper is very well-written, and it is easy to read it. However, the authors might want to consider the following comments to improve the work.

-          The Discussion and Conclusion sections need improving. It is not very clear how the paper answered the research question and based on what methodology and how they validated the outcome.

-          In the discussion section, there is a need to review the limitation of the proposed method. For example, is it possible to provide hands on experience for all the students? The cost implication of that? Other resource restrictions such as challenges on codelivery of one module by more than one teacher?

-          The case study is written and explained very well with detailed information. However, I feel there is a need to clearly link the methodology and case study to what has been concluded in the paper.

Author Response

  • The research questions, discussion and conclusion were revised. The methodology section was also added to validate the outcome.
  • The limitations of the proposed method were reviewed.
  • The concept of codelivery is mainly being recommended only for modules/subjects that require hands-on activities.

Reviewer 3 Report

I found this manuscript to be very interesting. I think the approach could be replicated elsewhere on the continent.  There is information that would be very useful to practitioners and students.  

Author Response

We appreciate the reviewer’s comment. Thank you. 

The methodology section was added.

Round 2

Reviewer 1 Report

You need to make a big effort in the paper to establish a relation between Research questions, observations (as it is your method) and results. Then in discussion section you can discuss on the research questions or in the proposed curriculum or the rest of the aspects, always demonstratin…what , why, how….

You have to go deeper…..I try to explain it in the next lines (I did not manage to attach a file….)

Abstract.

Observation  methods  through  video  recordings  and  onsite  were used to observe and analyze how the learners were interacting throughout the workshop. Based on  the  results,  it  was  concluded  that  this  approach could be  applied  in  classrooms  to  enable  the primary school learners apply computational and design thinking in preparation of becoming the producers and not only the consumers of the 4IR technologies.

Research Problem

Namibia has recently taken a step to catch up with the 4IR technology by assessing the country’s readiness for the fourth industrial revolution and this involved discussions on upgrading the country’s education to education 4.0. Education 4.0 focuses on motivating and exciting teaching and learning pedagogies using the emerging technologies such as robotics. Research on educational robotics in Namibia is at a very early stage and still needs more  exploration  hence  this  study  focused  on  getting  an  answer  to the  research questions below.

RQ1: How  can  a  participant-oriented  educational  robotics  workshop prepare primary school learners to apply 4IR technologies?

RQ2: How can education practitioners incorporate  robotics,  computational  and  design thinking in the Namibian curricular?

Methodology: Design Science  Research (DRS) Methodology, a  methodological  approach  that focuses  on  the  designing and development of innovative artifacts that solve societal problems. The artifacts in this context are the methods and the ICDC framework (see Table4.) used in the study to understand and apply the basics of robotics alongside computational thinking and design thinking.

**** Explain better the methodology.

In the abstract you say:

Observation  methods  through  video  recordings  and  onsite  were used to observe and analyze how the learners were interacting throughout the workshop.

¿What do you observe?

*** explain that more precisely, because is the basis for making any conclusión

*** During the section 4 (Results) you should describe the observations in order to be used for the conclusions. Actually is a kind of report of the educational experience, but the observations, or the criteria or aspects to be observed, etc…are not at all clear…..

**** In section 4, Results, for every subsection, I would expect something like:

4.1

a) We have designed the artifact/the activity….what ever (for example “Workshop Introduction”)

b) In the interaction with learners we have observed: the learners were insterested in Robotics….

c) We can conclude that (what, why, how)….……and this is related to RQ1 (what, why, how)….and to RQ2……(what, why, how)

Discussion

In this study, primary school learners with no-prior programming and robotics skills were equipped with not only the basics of robotics and programming skills but also with the 21st century skills which are collaboration,communication,creativity,critical thinking, design thinking and computational thinking.

******What do you mean?

The latter can be achieved through educational robotics workshops such as the one conducted at Nakayale Private Academy.

*****Why, how, relate that to the results, RQ1, RQ2, Observations…..

integration of educational  robotics  and  computational  thinking  into  the  curriculum  of  both  primary  and secondary school level

****Explain that….whay, how?

*****And so on with the rest of the Points of the discussion…….

Reviewer 2 Report

NA!

Author Response

Thank you.